# Waste Generation Modeling Using System Dynamics with Seasonal and Educational Considerations

**Sanaalsadat Eslami** [1], **Golam Kabir** [1,*] and **Kelvin Tsun Wai Ng** [2]

[1] Industrial Systems Engineering, Faculty of Engineering and Applied Science, University of Regina, 3737 Wascana Parkway, Regina, SK S4S 0A2, Canada; sanaeslami1994@gmail.com

[2] Environmental Systems Engineering, Faculty of Engineering and Applied Science, University of Regina, 3737 Wascana Parkway, Regina, SK S4S 0A2, Canada; kelvin.ng@uregina.ca

*  Correspondence: golam.kabir@uregina.ca

**Abstract:** Effective waste management is critical to environmental sustainability and public health. Various dynamics, such as seasonal changes and waste education programs, influence solid waste generation, increasing the complexity of prediction. This is important, as the proper prediction of waste quantity is necessary to develop a sustainable waste management system. In this study, municipal solid waste (MSW) management is examined in Regina, the capital city of Saskatchewan, Canada. A system dynamics (SD) model is developed to evaluate garbage and recyclable waste generation behaviours in Regina across four seasons. Three years of Regina landfill waste generation records (2016–2018) are considered to analyze and predict seasonal waste-generation trends. The effect of various factors, such as gross domestic product (GDP), population, and education attainment on the amount of waste generation is considered in the SD model. The SD model is designed as a stock-flow diagram to illustrate the relationships between variables and predict the next three years of waste trends. This finding highlights the importance of waste education and awareness program and seasonal effects on the accuracy of SD waste modeling.

**Keywords:** municipal solid waste management; system dynamics; seasonal variation; recyclable waste; education and awareness; recycling behaviors

## 1. Introduction

The rapid socioeconomic development that has occurred in many urban centers and regions over the last few decades has led proper management of municipal solid waste (MSW) to receive a significant increase in attention [1,2]. According to the World Bank [3], 2.1 billion metric tonnes of waste are generated annually worldwide—a figure that, under current conditions, is projected to rise by 70%. Prediction and estimation of MSW generation using regression analysis, material flow models, and machine learning approaches have been reported [4–7]. Recently, several studies have used system dynamics (SD) modelling to analyze MSW management systems (e.g., [1,2,7,8]).

Most researchers that have employed SD modelling to analyze MSW management have considered population, gross domestic product (GDP), and waste volume [9,10]. Wang et al. [8] designed various SD models to evaluate the relationship between socioeconomic benefits and waste separation in Tianjin, China. They found that a rise in waste separation is correlated with an improvement in socioeconomic benefits. Dianati et al. [11] designed SD models to assess waste volumes, including mixed waste and food waste, in Kisumu, Kenya. These models incorporated greenhouse gas emissions to assess the atmospheric impact of solid waste burning. Table A1 presents the summary of waste management studies using SD models.

Most waste studies have focused on a single type of waste to assess the effect of policies on specific types of waste treatment or disposal. For example, Dhanshyam et al. [1] used SD modelling to investigate various policies' effects on plastic waste generation

in India, considering socioeconomic level, recyclable rate, plastic waste generation rate, and waste sources. Some recent studies have assessed the impact of certain policies and scenarios on MSW management as a whole [11–13]. Such systematic approach may aid in minimizing global waste generation and their corresponding environmental impacts. Lu et al. [2] developed a SD model to predict total waste generation by considering GDP, population, local policies and conditions in China's Southern Tai Lake Watershed. The modeling results suggest that certain variables are more effective in extending the capacity of landfills for longer period of time. Such findings may enable governmental agencies and policymakers to better manipulate waste generation characteristics and recycling behaviors of the residents, improving overall sustainability.

Rafew et al. [12] estimated long term MSW quantity using a SD model in Khulna, Bangladesh. Their SD model considered several socioeconomic variables and looked at the entire life cycle of MSW management from waste collection to permanent land disposal through 2050. Ultimately, they found a need to increase the city's budget for both collection and landfill development.

Education and the awareness of residents is one policy element that can improve MSW management and environmental sustainability [10,14,15]. Due to the nature of this parameter, education is difficult to quantify precisely and thus mostly ignored in waste studies. The City of Regina, the capital city of Saskatchewan, Canada, recently identified education as a key factor in their MSW management strategies [16]. This education factor largely refers to knowledge and awareness of the residents on how to distinguish and properly dispose recyclable waste from non-recyclable waste. According to the City, educational efforts using various outreach program and recycling campaigns have led to a 16% decrease in waste generation between 2018 and 2020 [16]. As such, education as defined by the City is adopted in the present study to model MSW waste generation rates.

The proposed model also considers seasonal effects on MSW generation characteristics and recycling behaviors [17,18]. Knowledge of the climatic drivers behind MSW generation may aid policymakers in better responding waste quantity fluctuations, thereby mitigating health concerns during a global pandemic [19,20]. Seasonal effects on waste generation rate and composition appears site specific. For example, minimal seasonal variations on waste composition are reported in a Danish residual household waste study [21], whereas strong seasonal variations on food waste are reported in a Chinese study [22]. In this study, the objective is to develop an SD model to evaluate the effects of several socio-economical variables on waste generation rate, with the consideration of the education of the residents and seasonal variations. By examining the effects of education seasonal variations on the amount of waste generation, we may uncover means of improving MSW management processes. This is especially important in Canada, where the historical per-capita waste diversion rates were noticeably lower than other industrialized nations [23,24]. Canadians send the majority of their waste for permanent land disposal, contributing to the risks of landfill gas emission [25–27] and leachate groundwater contamination [28,29]. This SD model reported in this paper can, however, be generalized to other cities all over the world. The modeling results of this study reveal various ways in which decision-makers can advance the planning and operation of MSW management systems.

The paper Is organized as follows. Section 2 presents the methodology followed in this study. Results and outcomes of the study are analyzed and discussed in the following section. The model validation and scenario analysis are summarized in Section 5. Finally, the last section highlights the conclusions and limitations and suggests future research.

## 2. Methodology

MSW generation rates in Regina, Canada, have been reported in a number of studies [18–20], and it is selected as the study area. Population and economic growth in Regina over the last several decades have led to significant increases in waste disposal [16], prompting the city to consider alternative MSW management plans. Figure A1 presents the population and growth rate of Regina, SK, Canada. In this study, we examined the residents'

waste generation characteristics and recycling behaviors to predict future MSW generation using SD models. Regina covers 118.4 square kilometres, making it the province's second-largest city. The population of Regina, which has been growing for several decades, was 258,960 in 2020 [30]. Given the resultant rise in waste generation, Regina drew up the Official Community Plan, which proposed various methods of minimizing waste and, in turn, mitigating environmental issues. Socioeconomic factors, such as population and GDP, influence waste disposal [8,11]. Birth, death, and immigration rates are effective factors behind population, and the city's GDP has risen over the last decade, driving up waste generation [30].

Given the nature of the problem, we have adopted an SD modelling approach in this study to predict the amount of waste generation in Regina. SD models was developed in the 1960s to analyze large, complex systems with a non-linear model [31]. Causal loops are designed to illustrate an overview of the SD model's processes. SD models work with stocks and flows, with the relationships between stocks illustrating the root of the problem in the model. The stocks also show the main variables' processes separately, displaying the interconnected behaviour of the SD systems. SD models are appropriate for systems that work overtime and include various linguistic parameters and policies, as they better capture the effects of these variables. Literature has suggested that SD model performance is comparable or more accurate than other predictive models [2,11]. SD models are appropriate for assessing MSW management processes [8,12,13], and they provide insights into the effects of variables on waste generation.

### 2.1. Causal Loop of the SD Model

Figure 1 presents the causal loop of our MSW generation SD model using literature. Causal loops illustrate the positive and negative relationships between the variables, with the positive (+) and negative (−) flows indicating the increasing and decreasing effects of the cause-effect parameters, respectively. This specific casual loop illustrates the relationships among population, education, and GDP with respect to the nine types of waste.

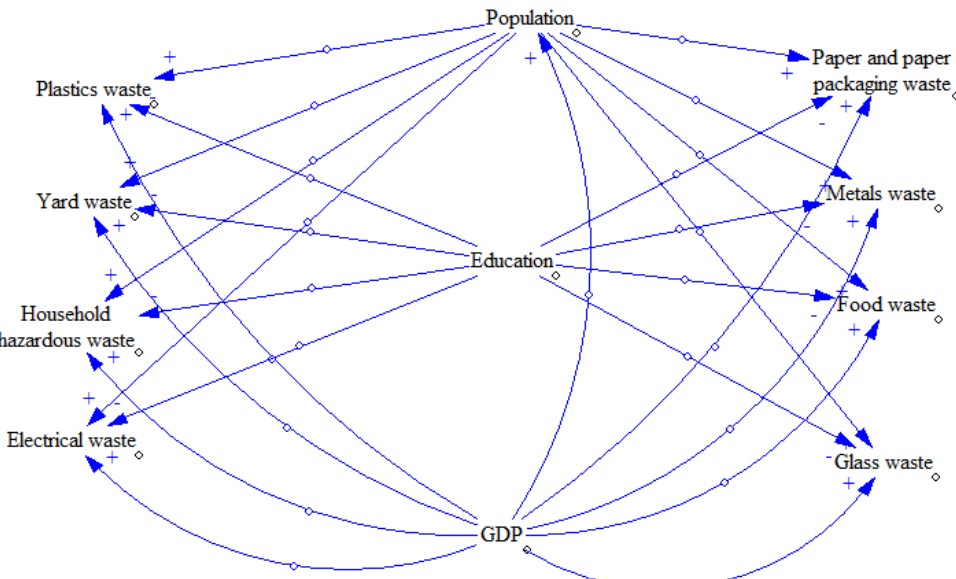

**Figure 1.** Casual loop diagram for the Regina municipal solid waste management system dynamics model.

As population and GDP grow, the waste generation rate increases, meaning they possess a positive relationship [8,13]. Liu et al. [15] investigated the influence of public education on residents' willingness to classify household waste. The performance of the education program was systematically evaluated and reported annually by the Taiyuan City of China [16]. The education parameter is expressed as a percentage with respect to the

residents' knowledge on their ability to identify non-recyclable wastes from recyclables. A higher percentage denotes a more effective and accessible waste education and awareness campaign. In this study, education or awareness programs have negative and positive effects on non-recyclable waste (garbage) and recyclables, respectively, with the term "garbage" referring specifically to the non-recyclable waste fraction.

*2.2. Stock-Flow Diagram and SD Model*

A recent waste study suggests that modeling by multiple waste streams is more advantageous [19] than the traditional approach, and therefore our SD model assesses the generation of nine different waste types (i.e., paper waste, paper packaging waste, food waste, plastic waste, glass waste, yard waste, household hazardous waste, electronic waste, and metal waste) in Regina. The proportions of garbage (non-recyclable waste) and recyclables were considered across four seasons (winter, spring, summer and fall). In this paper, stocks and flows were used to illustrate the city's waste generation processes and the relationships between policies and various waste types. In this study, Vensim PLE software was used to build the SD model. Figure 2 presents the stock-flow SD model for waste generation in Regina. The SD model was developed using three years (2016–2018) of Regina landfill waste records. The amount of garbage and recyclable waste generation were measured from brown garbage cart and blue recycling cart statistics in Regina. GDP and population constitute the model's socioeconomic variables, both of which have a direct effect on waste generation. The details of the variables are summarized in Table A2. The rates in the SD model illustrate the changes in waste generation volumes. The model's initial validity was assessed by examining the accuracy of its structure and its ability to replicate behavior, as compared to real-world observations [12]. In this study, the calibration process involves comparing the historical data on waste generation in the Regina landfill from 2016–2018 with the data generated by the model.

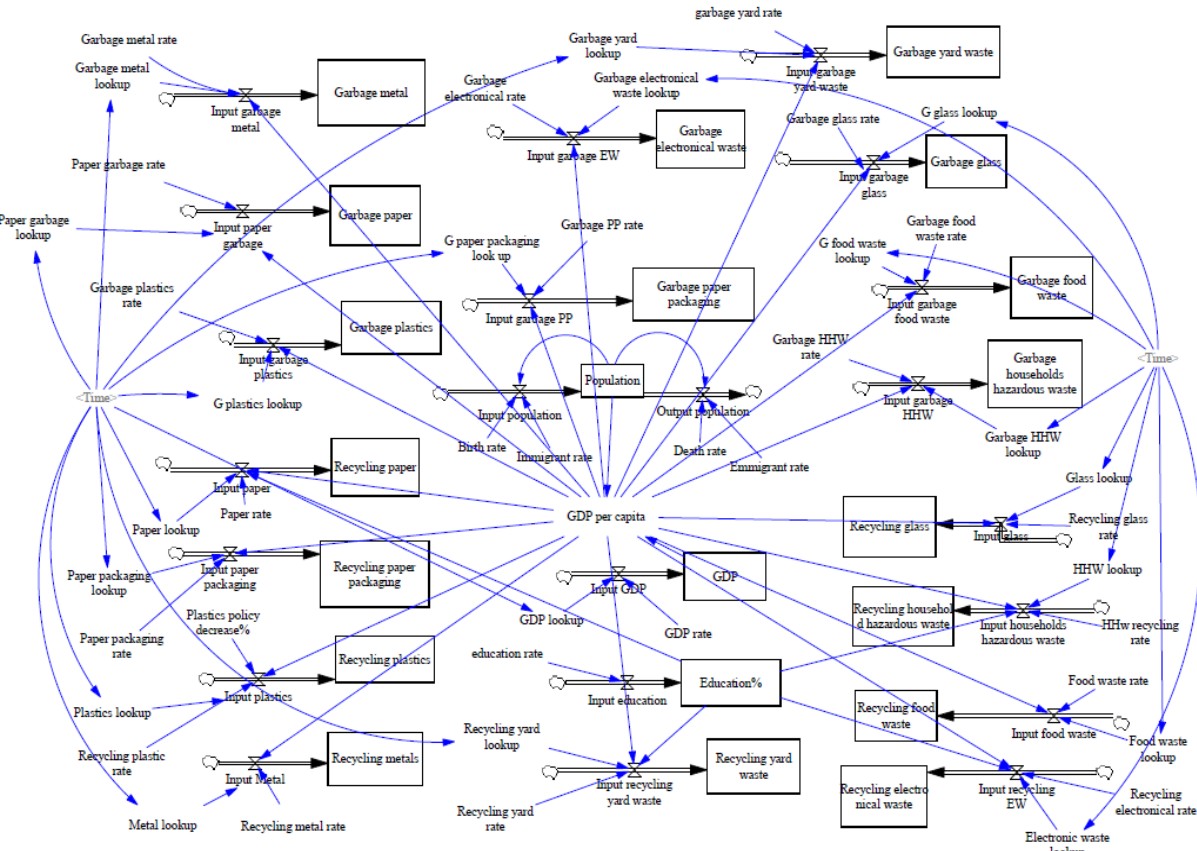

**Figure 2.** Regina system dynamics model for garbage and recyclable waste generation.

## 3. Results and Discussion

According to the historical statistics, Regina's GDP increases at an annual rate of about 1.5%; therefore, the SD model predicts that the GDP will increase from $16,194 M to $17,505 M between 2016 and 2021. Additionally, the model predicts a population increase from 344,751 to 357,836 between 2018 and 2021. Population growth stems from several variables, including births, deaths, and immigration and emigration rates.

The models predict the quantity (in million tonnes) of nine different waste types from 2019 to 2021 (Figures 3–7). The predicted waste trends stem from the effects of population, GDP, and education. Various waste-focused educational programs and awareness campaigns have been held to boost MSW management knowledge among the public in Regina, and data suggests that resident behaviors have been changing gradually during the study period [32]. For example, more consumers decided to choose paper bags over plastic bags and to select products with less plastic packaging due to the educational programs on waste minimization.

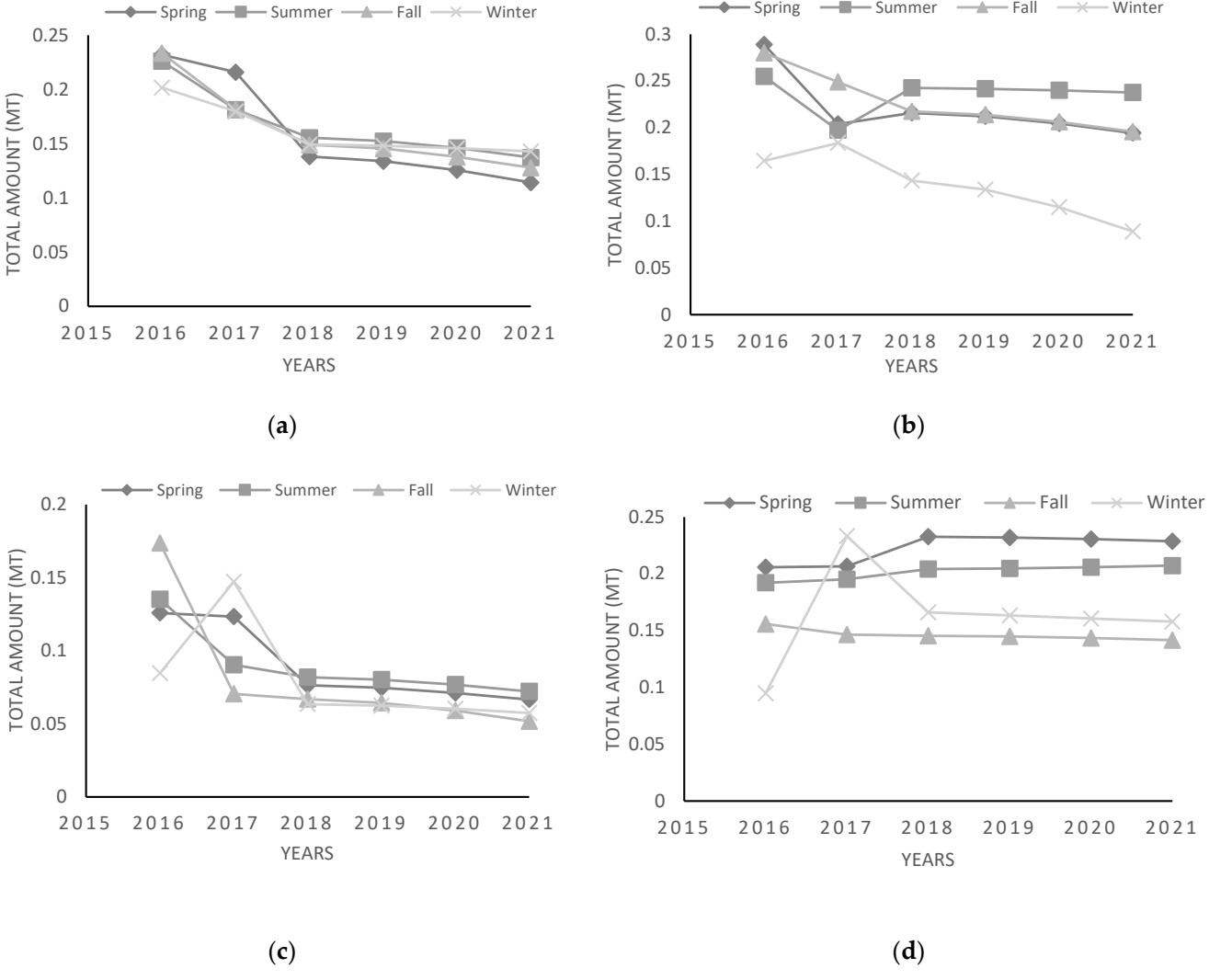

**Figure 3.** Waste prediction of (**a**) garbage paper, (**b**) recyclable paper, (**c**) garbage paper packaging, and (**d**) recyclable paper packaging.

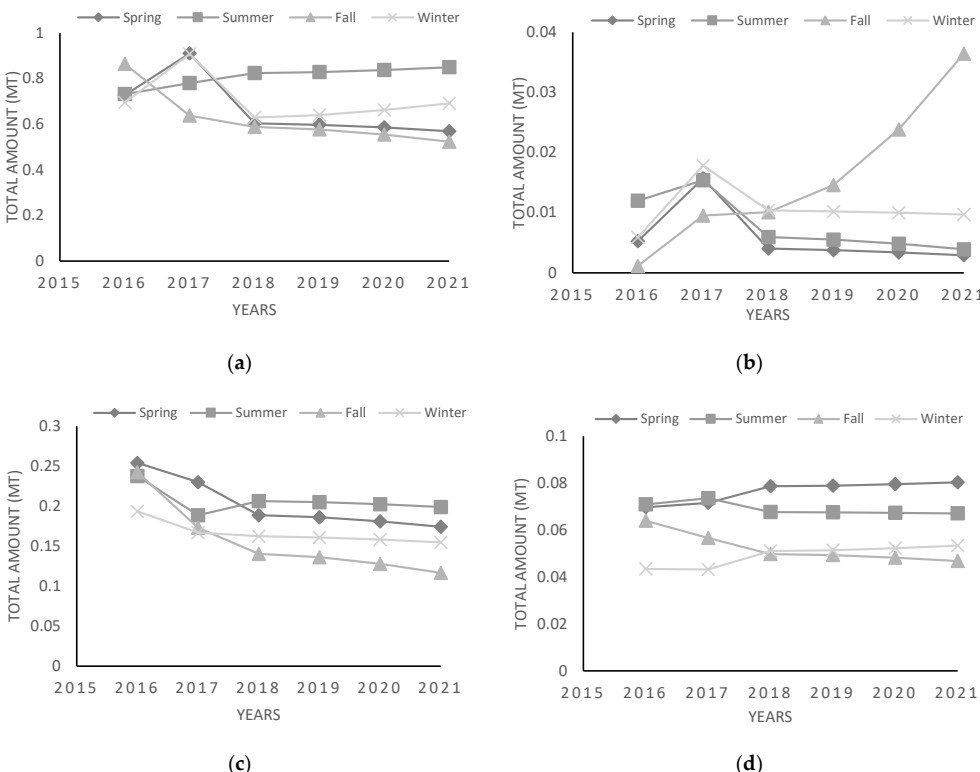

**Figure 4.** Waste prediction of (**a**) garbage food, (**b**) recyclable food, (**c**) garbage plastic, and (**d**) recyclable plastic.

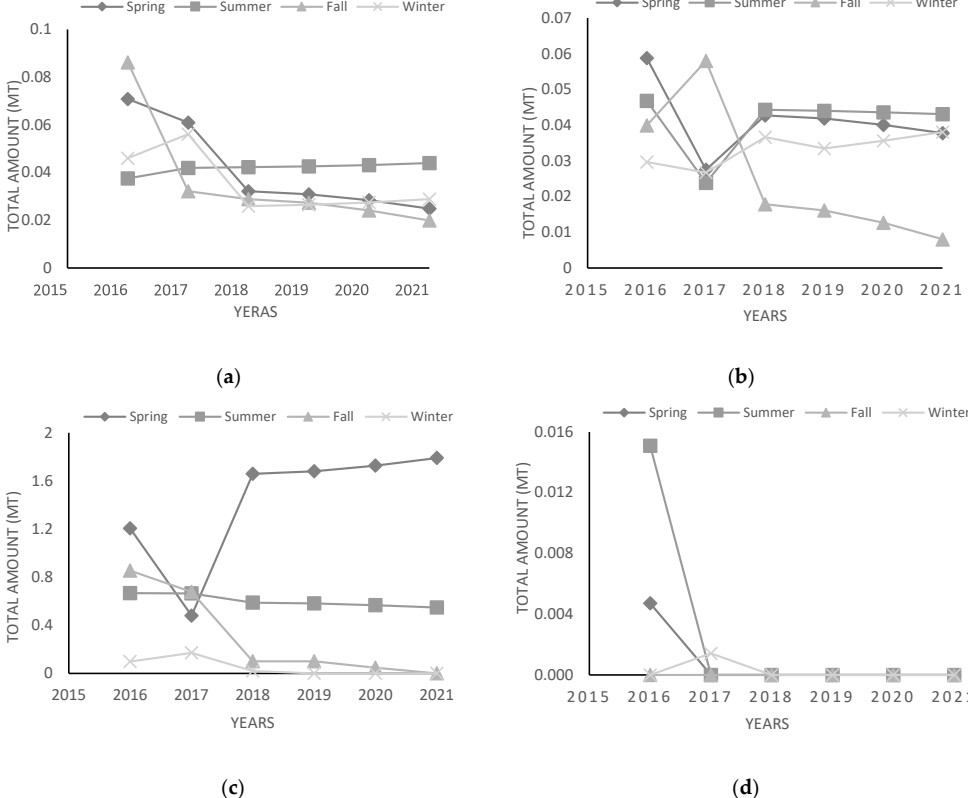

**Figure 5.** Waste prediction of (**a**) garbage glass, (**b**) recyclable glass, (**c**) garbage yard waste, and (**d**) recyclable yard.

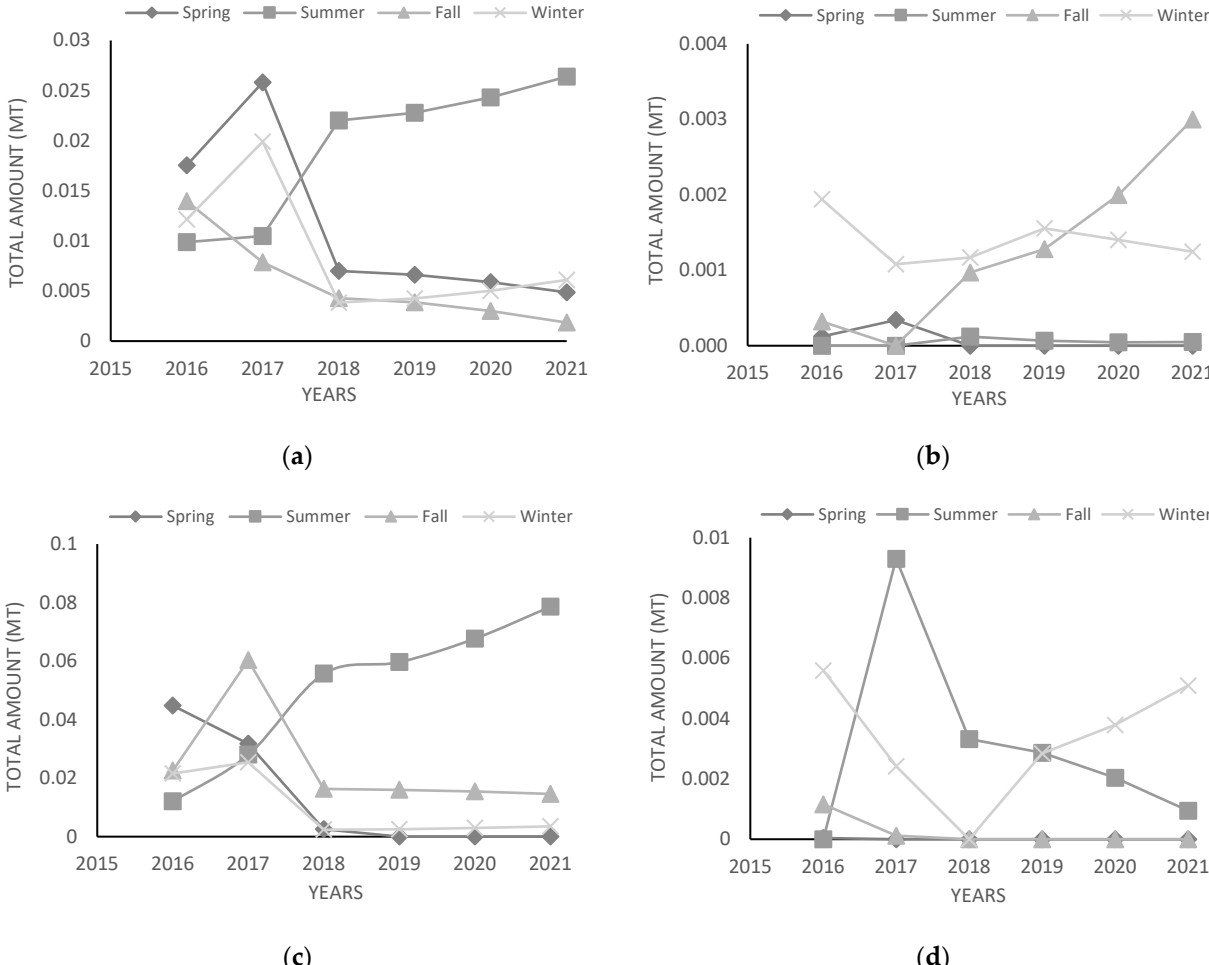

**Figure 6.** Waste prediction of (**a**) garbage household hazardous waste, (**b**) recyclable household hazardous waste, (**c**) garbage electronic waste, and (**d**) recyclable electronic waste.

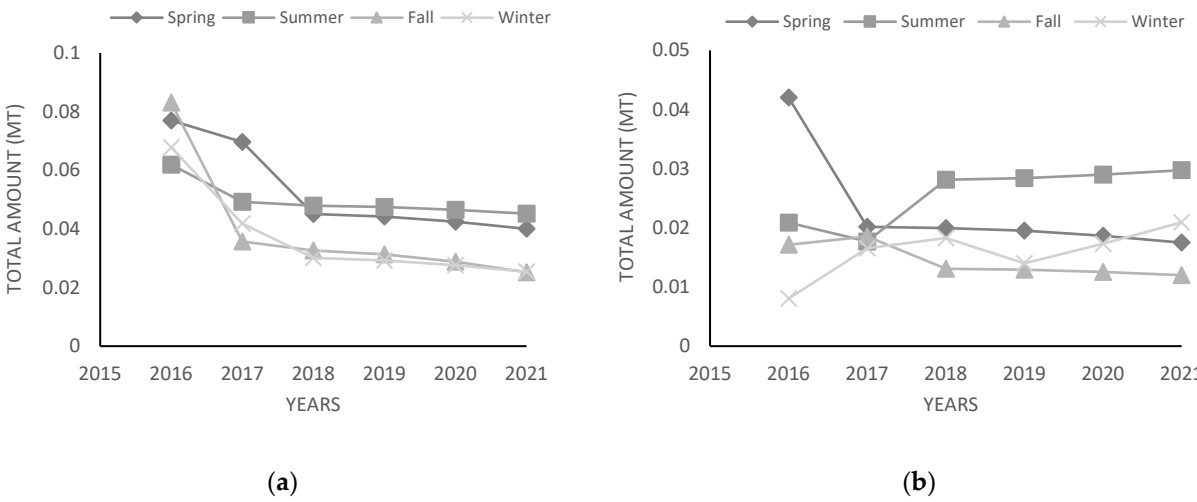

**Figure 7.** Waste prediction of (**a**) garbage metal and (**b**) recyclable metal.

Figures 3–7 illustrate the actual statistics (2016–2018) and predictions (2019–2021) on different waste fractions in Regina with respect to the four seasons. Figure 3a shows garbage (non-recyclable) paper waste generation rates, while Figure 3b shows recyclable paper waste generation rates. In general, the amount of recyclable paper waste exceeds that of garbage paper waste, indicating that the population has been appropriately separating

its paper. Between 2016 and 2018, the largest amount of garbage paper waste generation occurred during the spring (Figure 3a). However, there was a sharp decrease in paper waste generation between 2017 and 2018 as a result of promoting a culture of waste segregation in schools, showing the significant impact of education variables in this part. For this, the simulation predicts less garbage paper waste during the spring between 2019 and 2021. Figure 3b shows that the generation characteristics and recycling behaviors of recyclable paper is heavily influenced by the seasons, with noticeably lower waste generation rate in winters.

As shown in Figure 3c,d, the SD model predicts that the garbage paper packaging waste generation rate is greater during the summer than during other seasons, while the recyclable paper packaging waste generation rate is greatest during the spring, with approximately 0.25 metric tonnes between 2016 and 2021. The model predicts more recyclable paper packaging waste generation than garbage paper packaging waste generation. It appears that the seasonal effects are more important on the amount of recyclable paper and paper packaging (Figure 3b,d) than others. The waste predictions from 2019 to 2021 are quite consistent with a mild decreasing trend in all four waste fractions.

The SD model's predictions for garbage and recyclable food waste are shown In Figures 4a and 4b, respectively. There is a mild increase in garbage food waste generation during the summer and winter months between 2019 and 2021. Evidently, the highest amount of garbage food waste is generated during the summers, from 0.732 million tonnes in 2016 to 0.850 million tonnes in 2021. Li et al. [22] recently investigated the changes in food waste bulk densities between summer and winter in Beijing, China, and reported higher food waste bulk densities in summer. Figure 4b shows that the recyclable food waste generation rate is noticeably lower than garbage food waste generation. However, there has been a sharp increase in recyclable food waste generation during the fall from 0.001 million tonnes in 2016 to 0.036 million tonnes in 2021. This sharp increase is probably due to the increasing trend in the baseline period from 2016 to 2018 (Figure 4b). In general, due to the general increase in the amount of food waste generation between 2016 and 2018, the SD model predictions from 2019–2021 are quite consistent in the four seasons. In The City has recently implemented food- and yard-waste pilot programs to address the future changes of these wastes [33]. These projects may aid Regina in mitigating organic waste generation and achieving a higher waste diversion rate.

Garbage and recyclable plastic waste generation rates are plotted in Figures 4c,d, respectively. The general decreasing trend in the amount of plastic garbage waste is evident across all four seasons. The highest amount of plastic garbage waste generation occurs during the summer. On the other hand, the highest amount of recyclable plastic waste occurs during the springs, with about 0.002 million tonnes per year. In all cases, the predicted amounts for 2019–2021 are quite different among the seasons. Information on changes in waste generation characteristics and recycling behaviors across the seasons could provide interesting and valuable insights on planning and operating of a sustainable waste management system [19]. Many regulatory and administrative measures could be taken to mitigate plastic waste generation; for instance, Regina passed the Plastic Checkout Bag Ban Bylaw in July 2020. This is particularly important during a global pandemic, when higher plastic waste generation from masks, gloves, and other personal protective equipment is expected [34].

Figure 5 illustrates the predicted amount of garbage and recyclable glass and garbage and recyclable yard waste. Figure 5a shows a sharp decrease in garbage glass generation in fall and spring from 2016 to 2018, which affects the prediction values for the 2019–2021. Most glass waste in Regina consists of food and beverage containers, which can be reused to minimize the amount of glass waste generation. The summer features the highest volume of garbage and recyclable glass waste. Figure 5c,d show garbage and recyclable yard waste, respectively. As shown in Figure 5d, the quantity of recyclable yard waste in Regina is negligible (generally < 0.001 million tonnes) for 2017 and 2018. For this, the predicted values from 2019 to 2021 are also negligible. Garbage yard waste, however, is most prominent in

the spring and summer due to seasonal landscaping efforts and the pruning of trees and shrubs. This sharp decline in yard waste between 2017 and 2018 can be attributed to the opening of the city's yard waste depot [33].

Figure 6 illustrates the volume of garbage and recyclable household hazardous waste and garbage and recyclable electronic waste between 2016 and 2021. The prediction (between 2018 and 2021) in Figure 6a illustrate that most household garbage was generated during the summers because of the increasing trend in the baseline period from 2016 to 2018. Figure 6b highlights that the most household recyclable waste was generated during the falls and winters. Common household hazardous waste in Regina includes cleaners, batteries, solvents (e.g., acetone), and all poisonous materials [33]. Education can lead to a decline in the generation of household hazardous waste by raising awareness about recycling of this waste to minimize health risks and to mitigate environmental footprints.

Figure 6c,d show the volume of both recyclable and garbage electronic waste. Evidently, electronic waste generation peaks during the summer months. Electronic waste refers to electrical and electronic devices, including household appliances, machine tools, monitors, computers, and others. Tutton et al. [35] presented that there was a fluctuating trend for electrical waste in Canada due to the pre-processing of e-waste systems during 2016–2018. According to a Spanish study, environmental education is a key factor in improving electronic waste management [36].

The volumes of garbage and recyclable metal are shown in Figures 7a and 7b, respectively. There was a downward trend from 2016 to 2018 across all seasons for the quantity of garbage metal, which includes aluminium, steel cans and foil containers. For this reason, the predicted value of metal garbage during the summer and spring is around 0.05 million tonnes from 2019 to 2021, or about 0.02 tonnes higher than that during the fall and winter.

## 4. Model Validation and Simulation Scenarios

Promoting a culture of waste sorting and waste diversion is one key facet of the education variable in this research. An education variable is included in our SD models to assess the effects of educational program and awareness campaigns on waste generation behaviors in Regina. Due to the nature of the parameter, two scenarios considered specifically (with and without waste-focused education) to verify the sensitivity of the parameter on modelling results. Figure 8 illustrates the six recyclable generation waste quantities (paper, paper packaging, plastic, metal, glass, and food wastes) in Regina for the two scenarios. The SD models predict neglectable amount of household hazardous, yard, and electronic recyclable wastes between 2019 and 2021, and these three waste fractions are therefore ignored in this sensitivity analysis.

Figure 8a,b suggest that the educational campaigns are effective. Figure 5a shows that the amount of recyclable paper waste generation declined between 2019 and 2021 due to educational efforts. This effect of education is evident across all seasons.

Recyclable plastic waste is shown in Figure 8c. Most recyclable plastic waste generation has been generated during the spring and summer, which can be attributed to the high number of picnics and holidays during these seasons [33]. Evidently, more policies are necessary to minimize plastic waste, especially during the spring and summer. Figure 8c clearly visualizes the effect of education, as recyclable plastic waste generation is greater alongside educational efforts.

Recyclable glass waste generation and metal waste generation have increased between 2019 and 2021, as shown in Figures 8d and 8e, respectively. The effect of education can be clearly observed. For example, recyclable metal waste generation is at least 0.05 million tonnes higher with education than without education. This positive association with education is evident across all seasons.

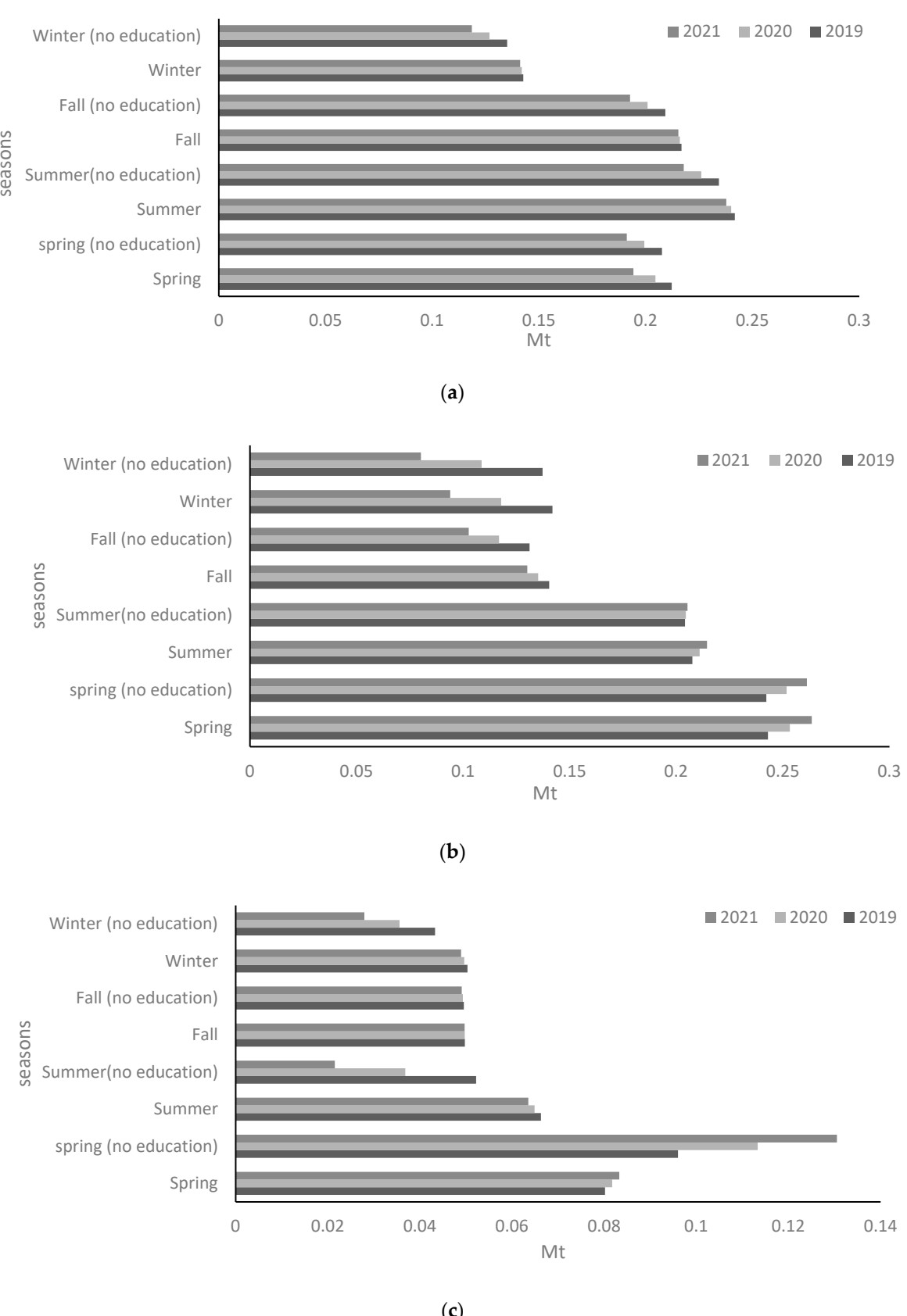

**Figure 8.** *Cont*.

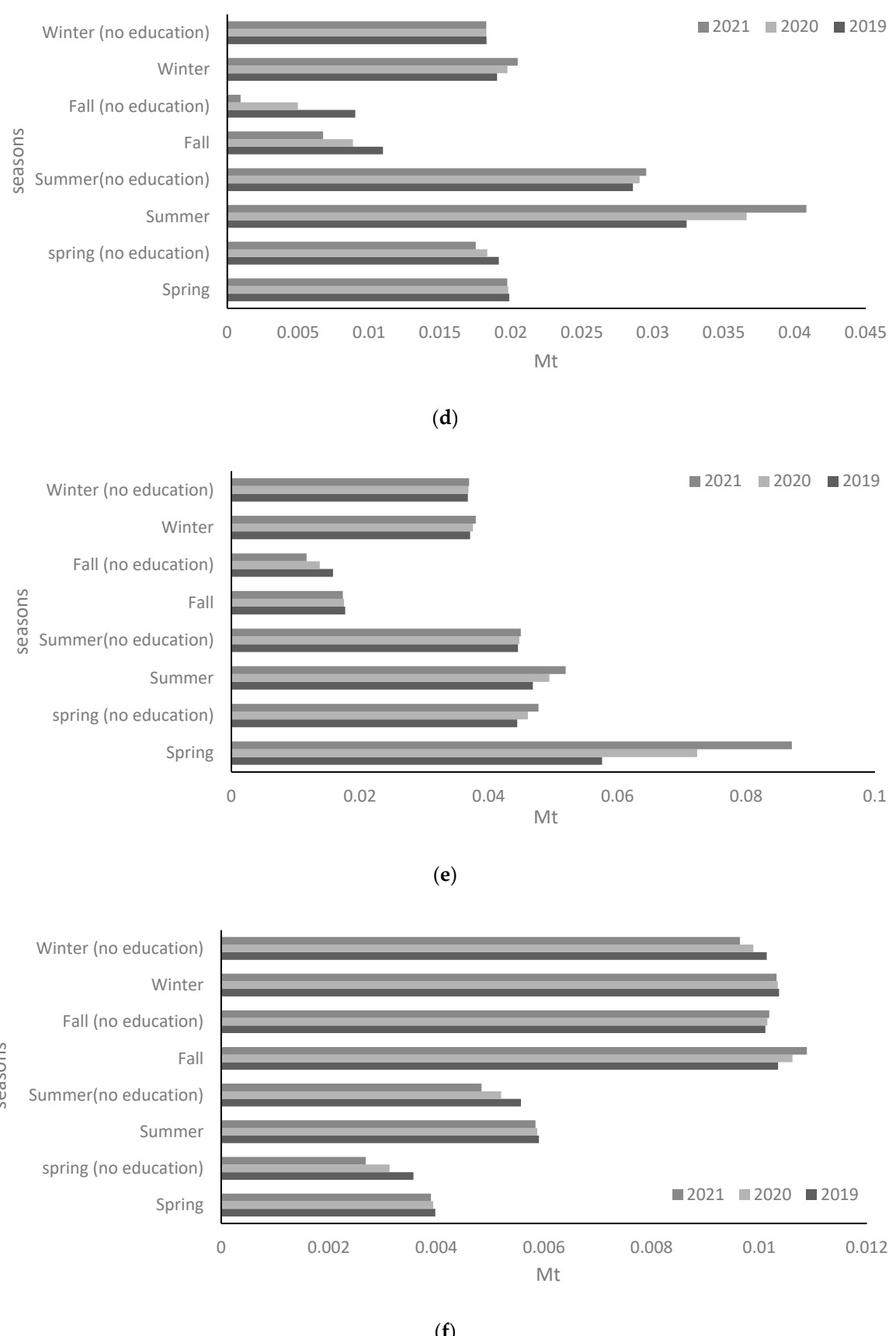

**Figure 8.** Recyclable waste simulation with and without the effect of education. (**a**) Paper; (**b**) Paper packaging; (**c**) Plastic; (**d**) Metal; (**e**) Glass; (**f**) Food waste.

Figure 8f illustrates recyclable food waste generation both with and without education. It demonstrates that the amount of recyclable food waste generation is predicted as 0.0104 to 0.0103 million tonnes with the effect of education; however, the same situation without the effect of education is slightly lower, at 0.0101 to 0.0096 metric tonnes between 2019 to 2021.

Evidently, education brings about a general increase in the amount of recyclable waste. This suggests that education on waste minimization and recycling can aid Regina in improving its waste diversion rate. Maddox et al. [37] investigated the effects of waste education programmes on residential waste in UK and found that both old and young audiences are important to the success of waste minimization and recycling initiatives. Similar findings are reported by Halkos and Petrou [38] using 20-year data from 25 OECD countries.

## 5. Conclusions

In this study, a SD model was designed to predict the volumes of different types of waste between 2019 and 2021 in Regina. Paper, paper packaging, food, plastic, metal, household hazardous, glass, yard and electronic waste were considered for distinguishing between garbage (non-recyclable) and recyclable waste. Three years of Regina landfill waste records were used as inputs and the effects of three socio-economical variables, namely GDP, population, and education, on MSW generation rates were assessed. Unlike other similar SD waste models, seasonal waste generation rates were explicitly modeled in this study, improving the overall model accuracy. Furthermore, education is a critical factor on waste diversion, as outlined in the Regina Waste Plan. The results showed that education plays a significant role in MSW management processes across all seasons and for all types of waste. The effects of seasons were found to be sensitive to waste type.

In addition, the most waste generation occurred during the summers, likely due to the high frequency of holidays and social gatherings in the warmer weather. In contrast, food garbage waste generation peaked during the winters and springs. Interestingly, there was a 41% reduction in paper garbage waste and a 26% reduction in plastic garbage waste across all seasons between 2016 and 2018. Therefore, the SD model predicted a declining trend on these waste streams.

In this study, the variables of education, GDP, and population were considered in the SD model. However, future research could certainly employ other variables related to the local MSW management policies and practices and consider more data for better prediction. Seasonal effects and education appear significant in waste modeling and are recommended in future studies. The suggested system dynamics have multiple applications. They can be employed for forecasting landfill capacity, examining long-term sustainable strategies, evaluating environmental quality, taking into account customer concerns, assessing employee performance, and analyzing the budget of a municipality. Furthermore, the suggested system dynamics can be combined with various multi-criteria decision-making methods to address the uncertainties and ambiguities involved in decision-making.

**Author Contributions:** Conceptualization, S.E., G.K. and K.T.W.N.; methodology, S.E., G.K. and K.T.W.N.; software, S.E., G.K. and K.T.W.N.; validation, S.E., G.K. and K.T.W.N.; formal analysis, S.E.; investigation, S.E., G.K. and K.T.W.N.; resources, G.K. and K.T.W.N.; data curation, S.E.; writing—original draft preparation, S.E.; writing—review and editing, G.K. and K.T.W.N.; visualization, S.E., G.K. and K.T.W.N.; supervision, G.K. and K.T.W.N.; project administration, G.K. and K.T.W.N. All authors have read and agreed to the published version of the manuscript.

**Funding:** The research reported in this study was partly supported by a grant from the Natural Sciences and Engineering Research Council of Canada (ALLRP 551383-20).

**Institutional Review Board Statement:** Not applicable.

**Informed Consent Statement:** Not applicable.

**Data Availability Statement:** Data may be available upon request.

**Acknowledgments:** The authors are grateful to the City of Regina Environmental Services branch for supporting this project. The views expressed herein are those of the writers and not necessarily those of our research and funding partners.

**Conflicts of Interest:** The authors declare no conflict of interest.

## Appendix A

**Table A1.** Summary of waste management studies using SD model.

| References | Description | Factor | Location |
|---|---|---|---|
| Wang et al. [8] | Developed five different SD models to investigate the effect of separated food waste rates and socioeconomic benefits on the amount of greenhouse emissions and saving lands. | Socioeconomic benefit factors, anaerobic digestion factor | Tianjin, China |
| Dianati et al. [11] | Estimates the greenhouse gas (GHG) and PM2.5 in Kisumu, Kenya. | Waste collection, Biogas, Scattered waste | Kisumu, Kenya |
| Dhanshyam et al. [1] | The focus of this paper is about plastic waste generation and the objective is to use the effect of the policies on the amount of plastic waste generation. | Waste to energy, Illegal and legal production, Packaging factor, Recycling factor GDP | India |
| Lu et al. [2] | Investigates the waste generation processes using economic variables | GHG, GDP, Population, MSW landfilling, MSW composting, MSW incineration | Southern Tai Lake, China |
| Rafew et al. [12] | Estimates waste generation using SD model with considering social and economic factors. | Society concern, Composting Capacity, Landfill capacity, Required fund | Khulna, Bangladesh |
| Chica-Morales et al. [10] | Analyzed the amount of waste generation by using different policies such as education. | Education, Budget, Funds | Darkhan (Mongolia) |
| Xiao et al. [13] | Investigates the amount of unsorted and sorted MSW. | GDP, Population | Shanghai |
| Ding et al. [39] | Designed a SD model to reduce construction waste by using construction and design stage policies. | Waste reduction | China |
| Zulkipli et al. [40] | Used SD model for a waste generation without any specific policies. | Economic factor | Malaysia |

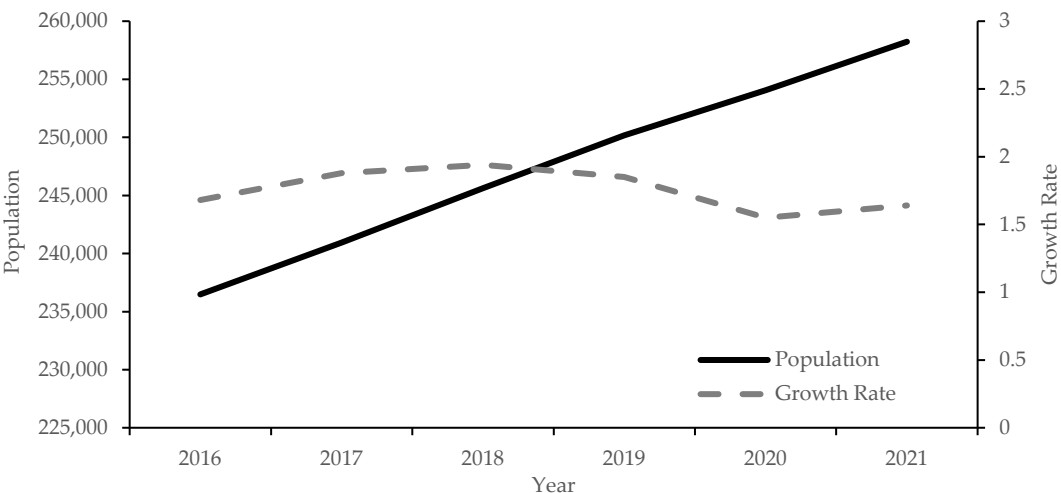

**Figure A1.** Population and growth rate of Regina, SK, Canada.

**Table A2.** Detail of the variables.

| Variable | Note | Value | Unit | Source |
|---|---|---|---|---|
| Population | Stock variable | 320,000 | person | https://www.canadapopulation.net/regina-population/ (accessed on 11 May 2023) |
| Input population | Flow, Population × (Immigrant rate + Birth rate) | | | |
| Output population | Flow, Population × (Death rate + Emigrant rate) | | | |
| GDP | Stock variable | 16,194 | $million | https://economicdevelopmentregina.com/economic-data/economic-report-card (accessed on 11 May 2023) |
| Input GDP | GDP rate × GDP lookup | | | |
| Paper garbage | Stock variable | Spring: 0.13802; Summer: 0.1556; Fall: 0.14909; Winter: 0.14927 | Tonnes | Regina landfill data |
| Paper-packaging garbage | Stock variable | Spring: 0.0763; Summer: 0.08187; Fall: 0.06694; Winter: 0.0636 | Tonnes | Regina landfill data |
| Metal garbage | Stock variable | Spring: 0.04509; Summer: 0.04796; Fall: 0.03266; Winter: 0.03009 | Tonnes | Regina landfill data |
| Glass garbage | Stock variable | Spring: 0.03221; Summer: 0.04229; Fall: 0.02885; Winter: 0.02593 | Tonnes | Regina landfill data |
| Household hazardous garbage | Stock variable | Spring: 0.007; Summer: 0.02203; Fall: 0.0043; Winter: 0.00385 | Tonnes | Regina landfill data |
| Plastics garbage | Stock variable | Spring: 0.188781; Summer: 0.20636; Fall: 0.14044; Winter: 0.162402 | Tonnes | Regina landfill data |
| Yard garbage | Stock variable | Spring: 1.657984; Summer: 0.58898; Fall: 0.10094; Winter: 0.01885 | Tonnes | Regina landfill data |
| Food garbage | Stock variable | Spring: 0.6041; Summer: 0.82474; Fall: 0.5886; Winter: 0.62993 | Tonnes | Regina landfill data |
| Electronic garbage | Stock variable | Spring: 0.00251; Summer: 0.05565; Fall: 0.01632; Winter: 0.00236 | Tonnes | Regina landfill data |
| Recyclable paper waste | Stock variable | Spring: 0.21596; Summer: 0.24263; Fall: 0.2175; Winter: 0.14342 | Tonnes | Regina landfill data |
| Recyclable paper-packaging | Stock variable | Spring: 0.232652; Summer: 0.20411; Fall: 0.14556; Winter: 0.16609 | Tonnes | Regina landfill data |



**Table A2.** *Cont.*

| Variable | Note | Value | Unit | Source |
|----------|------|-------|------|--------|
| Recyclable metal waste | Stock variable | Spring: 0.01995; Summer: 0.02812; Fall: 0.0131; Winter: 0.01828 | Tonnes | Regina landfill data |
| Recyclable glass waste | Stock variable | Spring: 0.04276; Summer: 0.0443; Fall: 0.01788; Winter: 0.03665 | Tonnes | Regina landfill data |
| Recyclable Household hazardous waste | Stock variable | Spring: 0; Summer: 0.00012; Fall: 0.00097; Winter: 0.00117 | Tonnes | Regina landfill data |
| Recyclable plastics waste | Stock variable | Spring: 0.07862; Summer: 0.06768; Fall: 0.04978; Winter: 0.051017 | Tonnes | Regina landfill data |
| Recyclable yard waste | Stock variable | Spring: 0; Summer: 0; Fall: 0; Winter: 0 | Tonnes | Regina landfill data |
| Recyclable food waste | Stock variable | Spring: 0.00402; Summer: 0.00594; Fall: 0.01008; Winter: 0.01039 | Tonnes | Regina landfill data |
| Recyclable electronic waste | Stock variable | Spring: 0; Summer: 0.00332; Fall: 0; Winter: 0 | Tonnes | Regina landfill data |
| Rates | Auxiliary variable | Calculated from the Regina landfill data | | Regina landfill data |

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
