# Peer review of "Waste Generation Modeling Using System Dynamics with Seasonal and Educational Considerations"

_sustainability, doi:10.3390/su15139995_

Round 1

Reviewer 1 Report

I have attached my questions in a pdf file

Author Response

Response: The authors would like to thank the reviewer for the critical and encouraging comments to improve the quality of the paper. We addressed all of your comments. Please check the attached file.

Reviewer 2 Report

The authors of the paper “Waste Generation Modeling” have developed a system dynamics model to predict future waste trends in Regina, Saskatchewan.  The model uses waste data from three years (2016-2018) while incorporating the impact of GDP, population and environmental education to predict seasonal trends for 9 types of municipal waste over the course of three years (2019-2021). 

My comments are as follows

1. Clarity

·         Line 18 – Remove “various socioeconomic factors, such as”.  Population is not a socioeconomic factor

·         Does “education attainment” refer to an individual’s level of education (e.g. high school vs college degree) or whether they have received education about waste and recycling.  

·         Line 21 – Replace “findings” with “finding”

·         Line 27-29 – The first sentence is unclear particularly “has led proper management of municipal solid waste to receive a significant uptick in attention”

·         Line 29-30  Clarify whether the “2.1 billion metric tonnes of waste” refers to Canada or global?

·         Tables 3-7 Would it be possible to delineate what is based on data (years 2016-2018) from what is modeled (years 2019-2021)?

·         Pages 11-13 –  Figure 8 needs a label.  

2. Methods

·         You mention that the City reports annually on the impact of its education program.  Could you provide more explanation on how you estimated the impact of education, particularly the nature of the source data?  E.g. How is the evaluation done? Does the City measure behavioral change, changes in the waste stream, or the number of participants?  This is important especially with the focus on the positive impact of the education programs.

3. Interpretation

·         Results lines 243-247.  Garbage electronic waste peaks during the summer months based on 2018 data and 2019-2021 predictions.  Did something happen between 2017-2018 to explain the change?

·         Results e.g. lines 198-199 “This sharp increase is probably due to the increasing trend in the baseline period from 2016 to 2019”.  Here and throughout the Results section, no distinction is made between data (years 2016-2018) vs modelled results (years 2019-2021). 

·         Abstract & Conclusion – Does the systems dynamics model have applications beyond this paper? Both the conclusion and abstract say that the model shows that waste education and seasonal effects are important.  What about the utility of the model approach itself?  Could it be applied to other years in Regina or used by other cities? 

No additional comments

Author Response

(The authors gave the same response as above.)

Reviewer 3 Report

This article uses Waste Generation Modeling that includes Seasonal and Educational factors to predict the landfill waste generation trend in Regina in 2016-2018 and 2017-2021, this study is interesting and valuable. Before publishing, address the following questions:

1. To make this work more attractive, it is recommended to predict more years, such as garbage trends after 2023.

2. The data points in figure3 should have error bar.

3. What is the software of the article model? Is it commercial software? Or an open source program? How are the parameters of the software set? If your own program, please open the program code, for others to check.

Quality of English Language is good

Author Response

(The authors gave the same response as above.)

Round 2

Reviewer 1 Report

Most of my previous comments were addressed. However, I don't see that the authors addressed this comment (9- Please include all equations for stocks as well as for flows.) 

Author Response

Most of my previous comments were addressed. However, I don't see that the authors addressed this comment (9- Please include all equations for stocks as well as for flows.) 

Response: The authors would like to thank the reviewer for the encouraging comment. We believe all the equations were defined in Table A2: Detail of the variables. Please note that the majority of the variables are stock variables and equations are not necessary for these varibles as we used the data from 2026-2018. Based on the Figure 1: Casual loop diagram for the Regina municipal solid waste management system dynamics model, we only need the equations for Input population, Output population, and Input GDP variables which are included in Table A2.
